# The development and validation of the Videogaming Motives Questionnaire (VMQ)

**Francisco J. López-Fernández**[1], **Laura Mezquita**[1,2], **Mark D. Griffiths**[3], **Generós Ortet**[1,2], **Manuel I. Ibáñez**[1,2]*

**1** Department of Basic and Clinical Psychology and Psychobiology, Universitat Jaume I, Castelló, Spain,
**2** Instituto de Salud Carlos III, Centro de Investigación Biomédica en Red de Salud Mental (CIBERSAM), Castelló, Spain, **3** International Gaming Research Unit, Psychology Department, Nottingham Trent University, Nottingham, United Kingdom

* iribes@uji.es

**Data Availability Statement:** All relevant data are within the manuscript and its Supporting Information files.

## Abstract

Gaming motives are important factors for explaining individual differences in videogame-related behaviors. The aim of the present study was to develop a new comprehensive but brief instrument–the Videogaming Motives Questionnaire (VMQ)–which embraces some of the most relevant gaming motives. In a first study, a pilot exploratory factor analysis (EFA) with data from 140 undergraduates was performed on items from twelve potential motives. This identified eight main factors: recreation, social interaction, coping, violent reward, fantasy, cognitive development, customization, and competition. In Studies 2 and 3, an EFA and a confirmatory factor analysis were performed on two independent samples of 407 adolescents and 260 young adults, respectively. The VMQ presented a robust eight-factor structure, with all scales showing adequate reliability indices. In reference to criterion validity, all motives presented specific associations with hours spent playing videogames, disordered gaming, and game genre preferences. More specifically, and in both adolescents and young adults, social interaction was the main motive related to time spent gaming, whereas disordered gaming was related to both coping and social interaction motives. Based on these findings, it is concluded that the VMQ is a brief and psychometrically appropriate tool for assessing the most relevant videogaming motives.

## Introduction

In the past three decades, videogame playing has become one of the most popular leisure activities worldwide [1]. In Spain (where the present study was carried out), 13 million people play videogames weekly and the highest rates of use are among adolescents (12–17 years) and young adults (18–24 year) [2]. Although most gamers play without any problems, there has been concern about the consequences of excessive use among a minority of individuals. This led to the introduction of Internet Gaming Disorder (IGD) in Section 3 ("Conditions for Further Study") of the latest (fifth) edition of the *Diagnostic and Statistical Manual of Mental Disorders*, DSM-5, defined as a "*persistent and recurrent use of the Internet to engage in games, often with other players, leading to clinically significant impairment or distress. . .*" [3]. In

**Funding:** The author(s) disclose receiving the following financial support for the research, authorship, and/or for the publication of this article: RTI2018-099800-B-I00 from the Spanish Ministry of Science, Innovation and Universities (MICIU/FEDER); GV/2016/158 and AICO/2019/197 from the Valencian Regional Government; UJI-A2017-18, UJI-B2017-74, UJI-A2019-08 and E-2018-16 from the Universitat Jaume I.

**Competing interests:** The authors have declared that no competing interests exist.

addition, the World Health Organization [4] included Gaming Disorder in the latest (eleventh) edition of the International Classification of Diseases (ICD-11) in the section "Disorders due to substance use or addictive behaviors".

These inclusions in the DSM-5 and ICD-11 assume that disordered gaming (for simplicity, this term will be used throughout the manuscript to describe the syndrome) shares a common symptomatology with substance use and gambling disorders [5]. The worldwide prevalence rates among representative samples range between 1 and 9% depending on age groups and sociocultural differences [6], with the highest prevalence rates observed at younger ages and among males [7]. In Spain, the prevalence of adolescent disordered gaming is around 7–8% [8–10]. Disordered gaming behavior can result in significant clinical impairment in the functioning of life areas (e.g., relationships, education and/or occupation, and psychological well-being) [7,11,12]. Furthermore, some studies have demonstrated that disordered gaming predicts internalizing and externalizing problems as well as poor academic performance over time [13–17].

Although structural and situational characteristics are involved in the acquisition, development, and maintenance of disordered gaming [18,19], personal variables of the gamers are also important, including motives for playing. Some models that contemplate the interrelated role of biological, psychological and social variables in the development of regular and disordered gaming consider motives as important predisposition components in gaming behavior [20,21]. Motives have shown to be especially relevant in other addictive-related behaviors, such as in the use and abuse of alcohol [22,23], marijuana [24,25] and gambling [26,27].

In the past few decades, different gaming motives questionnaires have been developed following two main approaches. From a more theoretical approach, research has attempted to identify relevant gaming motives based on well-established theories, such as *self-determination theory* [28], *uses and gratification theory* [29] and *social cognitive theory* [30]. For instance, based on self-determination theory, Ryan, Rigby and Przybylski [31] postulated that players would seek to satisfy universal psychological needs, such as autonomy, competence, relatedness, and presence. On the other hand, Sherry et al. [32] focused on why people use videogames to satisfy their psychological and social needs based on uses and gratification theory. Consequently, the scale developed by the authors comprised six motivational dimensions: arousal, competition, challenge, social interaction, recreation, and fantasy. More recently, De Grove, Cauberghe and Van Looy [33] developed a motives scale for digital gaming based on the social cognitive theory. The main tenet of this theory is that individuals are motivated to act as a function of outcome expectancies, and according to their study, they identified eight main motives for playing: social, narrative, escapism, agency, performance, pastime, moral self-reaction, and habit.

Following an empirical approach, a number of scales to assess videogame motives have been developed using exploratory procedures, such as factor analysis. For example, Yee [34,35] conducted two large studies among players of Massively Multiplayer Online Role-Playing Game (MMORPG) in order to identify their motivations. In the first study, Yee [34] described a hierarchical structure of motives, with three supra-factors and ten sub-factors: achievement (advancement, mechanics, and competition), social (socializing, relationship, and teamwork), and immersion (discovery, role-playing, customization, and escapism). In the second study, Yee [35] identified five main motives: relationship, immersion, escapism, achievement, and manipulation. Other researchers have extended the examination of motives in general. For instance, one of the motives scales most used for general videogame playing is the Motives for Online Gaming Questionnaire from Demetrovics et al. [36]. This scale, based on a literature review and players' interviews, posited seven global motives to play videogames: escape, coping, fantasy, skill development, recreation, competition, and social. Table 1 summarizes most

**Table 1. Comparison of the scales employed to develop the RPG scale according to the most relevant motives in video gaming research.**

| Motives | De Grove et al. [33] | Demetrovics et al. [36] | Ferguson & Olson, [78] | Floros & Siomos [85] | Fuster et al. [61] | Hilgard et al. [59] | Kahn, et al. [44] | Kim & Ross [86] | Lafrenière, Verner-Filion & Vallerand [87] | Lee & LaRose [88] | Li, Liau, Gentile, Khoo & Cheong [89] | Nackle, Bateman, Mandryk [90] | Nije et al. [60] | Olson [91] | Rodríguez de Sepúlveda & Igartua [92] | Ryan et al. [31] | Scharkow et al. [58] | Sherry et al. [32] | Tondello et al. [93] | Wallenius, et al. [94] | Westwood & Griffiths [95] | Wu, Wang & Tsai, [96] | Yee [34,35] & Yee et al. [45] |
|---|---|---|---|---|---|---|---|---|---|---|---|---|---|---|---|---|---|---|---|---|---|---|---|
| Social interaction | X | X | X | X | X | X | X | X | | | X | X | X | X | X | X | X | X | X | | X | X | X |
| Fantasy | X | X | | X | X | X | X | X | | | X | X | X | X | X | X | X | X | | X | X | X | X |
| Achievement-challenge | X | | X | X | X | | | | X | | X | X | X | | | X | X | X | X | | X | X | X |
| Competition | | X | X | X | | | X | X | | | | X | X | X | X | | X | X | X | | | | X |
| Recreation | X | X | X | | | | | X | | X | | | X | X | X | | | X | | | | X | |
| Escape-coping | X | X | X | X | | X | | | | | | | X | X | | | | | | X | | | X |
| Arousal | | | X | | | | | | X | | | X | X | X | X | | | X | | | | | |
| Customization | | | | | | X | | | | | | | | X | | | | | X | X | X | | X |
| Skill development | | X | | | | | X | | | | | X | | | | | | | | | | | |

of scales developed during the past decades as well as the most common motives identified in them.

One of the videogame genres in which motives have been more extensively studied is MMORPGs, mainly because of the high IGD prevalence reported in this genre [37–39]. Research examining MMORPGs has reported associations between gaming motives and disordered gaming (e.g., [34,40,41]), time spent gaming [35,42], types of positive and negative passion (or engagement) in gaming [43], and in-game behaviors [41,44,45]. Motives have also been related to different uses of other Massively Multiplayer Online games such as Massively Online Battle Arena (MOBA) games, and First-Person Shooter games [46,47]. Similarly, gaming motivations have been associated with disordered use in general gaming [47–53].

Other studies have demonstrated that some gaming motives mediate the relationships between psychiatric problems and disordered gaming (e.g., [52,54–56]). Additionally, motives have been associated with higher video gaming frequency [32,49,53,57], genre preferences [47,57,58] and favorite game franchises [59].

Taken together, these studies demonstrate the importance of motives in the psychological study of gaming behaviors, and the need to properly identify and assess them. However, the research to date presents some limitations. As shown, utilizing different theoretical background results in a different number and nature of motives. Furthermore, some of the described motives suffer from a form of 'jingle-jangle' fallacies, that is, that apparently similar constructs are sometimes given different labels (jangle fallacy) whereas the same labels are sometimes applied to conceptually different constructs (jingle fallacy). For instance, in relation to the fantasy component of the game, some scales highlight different aspects within the same motive, such as discovery and role-playing [34], fantasy, exploration and narration [58], and fantasy-escape and fantasy-arousal [60]. Conversely, different studies describe this component using distinct labels, such as immersion [34,45], presence [31], fantasy [32,36], narrative [33], story-driven [45], story [59], and exploration [61]. Finally, many studies have focused on specific games or genres, especially MMORPGs; e.g., [34,45,61], so it is unclear to what extent the proposed motives are specific to this genre, or common to all genres. This variability across the different gaming scales impedes the development of generalizable taxonomies of video game motives and hinder accurate communication in this field.

Therefore, and following an empirical approach, the main aim of the present study was to develop a new scale of gaming motives in which the most relevant and recurrent motivational components found in previous studies were reflected under unified labels that can be used for any gaming genre. To do so, three studies were carried out. The first study identified some of the most relevant motives through a literature-based scoping study and comprised an initial exploratory factor analysis (EFA) on items reflecting these motives in data collected from a sample of young adults. In Study 2, an EFA of selected items encompassing the videogame motives more clearly identified in Study 1 was conducted using data from an adolescent sample. Finally, in Study 3, a confirmatory factor analysis (CFA) was performed using data from a sample of young adults. The role of the identified motives was explored in relation to the number of hours spent gaming, disordered gaming, and game genre preferences in Studies 2 and 3.

## Study 1. Development of pilot scale and initial testing
### Method

**Participants and procedure.** For the development of a pilot scale and the initial selection of items, a convenience sample of 140 young adult players was used (76 undergraduates and 64 participants recruited from the LAN party, "UJI Game Experience", an event at the Jaume I University in which gamers gathered to play multiplayer video games; 79 males; mean

age = 20.71 years [*SD* = 3.55]). Participants voluntarily completed an online survey without any compensation via *Google Forms* comprising the pilot motive scale together with sociodemographic data.

**Measures.**   *Pilot Videogaming Motives Questionnaire (VMQ)*. Through an extensive review of the literature concerning gaming motivation scales, 25 relevant gaming motive scales were selected. For the review, a search was conducted in the PsycINFO and Google Scholar databases from 2000 to present, combining the key words motives or motivation, video game or gaming, and scale or questionnaire. Furthermore, citations from selected studies of our database were used to continue the review process until no other relevant study was found. The nine most frequently found motives in the literature were: arousal, recreation, social interaction, escape-coping, fantasy, skill development, customization, achievement-challenge, and competition. Two other motives–violence catharsis and violent reward–from Hilgard et al.'s scale [59] were also initially chosen given the potential relevance of violence and aggression in gaming [62]. Last, we also took into account the common motives for the consumption of different addictive substances, such as alcohol, tobacco and marijuana, i.e. enhancement/recreation, social, coping and conformity, so this last motive was also included [63]. A total of 62 items based on these motives were selected, with five items per motive except violent catharsis and social interaction (six items each) and adapted to Spanish from the original scales utilizing an internationally standardized back-translation process [64,65]. Participants had to answer each item using a five-point Likert scale, from 1 (*strongly disagree*) to 5 (*strongly agree*).

**Statistical analysis.**   The IBM SPSS Statistics V21.0 software was employed for the EFA, with principal axis factoring and oblimin rotation on the 62 items selected. For the parallel analyses, Monte Carlo PCA [66] was utilized. Although the sample size was below the expected norms in relation to the number of items factorized [67], the pilot study was only exploratory, and its main aim was to identify the clearer and more robust factors underlying the analyzed items.

**Ethics.**   Participants were adequately informed about the study, confidentiality and treatment of data, as well as the data protection procedure; and gave written informed consents. This study was approved by the ethical committee from the Universitat Jaume I and was carried out in accordance with the Declaration of Helsinki.

## Results

The eigenvalue criteria [68] suggested 12 factors whereas parallel analyses [67] suggested seven factors. Consequently, six EFAs were performed extracting from 7 to 12 factors. Across different EFAs, eight intercorrelated factors became more robust and well-defined (see S1 Table) explaining 78.7% of the variance: recreation, social interaction, coping-escape, violent reward, fantasy, customization, competition, and skill development. The other four motives were not well identified in the 12-factor solution: achievement-challenge, conformity, and violent catharsis were formed by only two items with high loadings; whereas the arousal items were distributed along different factors. In the eight-factor solution, the arousal items loaded principally on the recreation and fantasy motivations; violent catharsis items loaded on the escape-coping and violent reward dimension; conformity items loaded mainly on the social interaction dimension; and achievement-challenge items were distributed between different motives.

## Study 2. Adolescent sample study

### Method

**Participants and procedures.**   Adolescent participants were recruited from two Spanish high schools utilizing convenience sampling. Of the 1,106 students invited to participate, 835 adolescents returned signed written parental consent and received school supplies as an

incentive. From these, 407 reported a frequency of gaming of at least one hour weekly and completed the VMQ. Approximately two-thirds of the sample participants were male (68.2%) with a mean age of 14.99 years ($SD$ = 1.13).

This study was part of broader research into psychosocial risk and protective factors affecting mental health (see Moya-Higueras et al. [69] for more details). Trained research assistants administered the battery of questionnaires in two sessions, separated by approximately one week. A convenience sub-sample of 153 participants (75.2% boys; mean age = 15.08, $SD$ = 1.09) completed a third session one month later, in order to explore test-retest reliability.

**Measures.** *Videogaming Motives Questionnaire (VMQ)*. In order to develop a short instrument that includes scales with good reliability and adequate content validity, for each of the eight motives, three non-redundant items with high loadings and intercorrelations were selected and that adequately represented the content of the motive. Therefore, the 24-item definitive version of VMQ was used to assess eight gaming motivations: recreation, social interaction, coping, violent reward, fantasy, cognitive challenge, customization and competition using a five-point Likert scale (0 = strongly disagree; 4 = strongly agree). In relation to the coping-escape motive, the dimension became coping only because the three most representative items were exclusively related to coping aspects. Finally, the skill development motive became cognitive development because its most relevant items concerned mental challenge and development rather than improvement of abilities in general.

*Gaming behaviors*. Data were collected on the number of hours spent gaming daily online and offline per week (from Monday to Friday) and on weekends (Saturday and Sunday). Weekly number of gaming hours was obtained by adding online and offline daily hours and multiplying the days of these periods of the week by the total number of hours. Participants were also asked to provide up to five of their most played videogames. These games were then classified into the most popular videogame genres, based on previous studies [e.g., 9]. These gaming genres comprised: shooter games (e.g., *Call of Duty*), MOBA (e.g., *League of Legends*), strategy games (e.g., *Clash of Clans*), MMORPG (e.g., *World of Warcraft*), role-playing games (e.g., *Dark Souls*), action-adventure games (e.g., *The Legend of Zelda*), sport games (e.g., *FIFA*), casual games (e.g., *Candy Crush*), social simulation games (e.g., *The Sims*), construction games (e.g., *Minecraft*), platform games (e.g., *Super Mario Bros*), fighting games (e.g., *Street Fighter*) and other miscellaneous games.

*Disordered gaming*. A Spanish adaptation [9] of a measure especially developed to assess disordered gaming in adolescents [70] was employed. The 11-item scale is based on symptoms of addiction, such as withdrawal, conflict, tolerance, salience, mood modification and relapse. Participants indicate their frequency of videogame-related problems on a four-point Likert scale (from 0 = *"never or almost never"* to 3 *"almost always or always"*) during the past year. The Cronbach's alpha in the present study was very good (.85).

**Statistical analysis.** As the pilot study showed that motives were intercorrelated, an EFA with principal axis factoring and oblique rotation (oblimin) was performed for testing the VMQ with original eight-factor model using IBM SPSS Statistics V21.0. The Cronbach's alpha of each VMQ factor was calculated. Temporal stability reliability was computed utilizing a two-week test-retest correlation. In order to determine the criterion validity of the VMQ, the scores of the motives were correlated with number of weekly hours spent gaming, disordered gaming, gaming genre, age, and gender. Finally, hierarchical multiple regression analyses were carried out to examine the role of motives in hours spent gaming per week and disordered use. Previous research has shown that gender and age were relevant variables in video game–related behaviors, such as disordered use or video game genre preferences (e.g., [9,32,57]). Therefore, hierarchical multiple regressions were controlled for gender and age by introducing them in step 1, whereas motives were entered in step 2".

**Ethics.**    Parents or legal guardians of the adolescents were informed about the study, confidentiality, treatment of data, as well as the data protection procedure; and given written informed consents. The study was approved by the ethical committee from the Universitat Jaume I, and authorized by the school board of the participating high schools as well as by the Valencian regional education authorities, and has been carried out in accordance with the Declaration of Helsinki.

## Results

**Gaming behavior.**    Participants reported playing an average of 1.78 hours a day during the week (*SD* = 1.86), and four hours a day at the weekends (*SD* = 3.01). Therefore, the mean weekly time spent gaming was 16.89 hours (SD = 13.45). The most played videogames reported by participants (n = 359) were classified according to gaming genre: shooter (n = 197), sports (n = 150), MOBA (n = 71), strategy (n = 64), action-adventure (n = 45), social simulation (n = 35), casual (n = 33), role-playing (n = 23), platform (n = 23), construction (n = 21), fighting (n = 20), MMORPGs (4) and other miscellaneous games (n = 58).

**Factor structure.**    The EFA in the adolescent sample replicated the eight-factor solution found in Study 1 and explained 78.5% of the variance (see Table 2). The factor loadings for

**Table 2. Exploratory factor analysis and reliability of the VMQ.**

| Item | Factor Loading | Item | Factor Loading |
|---|---|---|---|
| **Recreation** | Alpha = .84 | **Social interaction** | Alpha = .79 |
| | Retest = .60** | | Retest = .78** |
| 1. Disfruto jugando. (*I enjoy gaming*). | .82 | 5. Hago nuevos amigos. (*I make new friends*). | .75 |
| 9.Me lo paso bien. (*I have fun*). | .78 | 13. Mediante el juego estoy en contacto con mis amigos. (*I keep in touch with my friends by gaming*). | .66 |
| 17. Es divertido. (*It is entertaining*). | .73 | 21. Así encajo en un grupo de gente que me gusta. (*Thus, I fit in with a group I like*). | .53 |
| **Competition** | Alpha = .76 | **Violent reward** | Alpha = .93 |
| | Retest = .67** | | Retest = .82** |
| 2. Me gusta ganar. (*I like to win*). | .55 | 6. Me gusta la violencia en el juego, cuanto más mejor. (*I like violence in video games, the more violent the better*). | .93 |
| 10. Me gusta demostrar que soy mejor que otros jugadores. (*I like to prove that I am better than other players*). | .55 | 14. En el juego es divertido disparar a alguien en la cabeza. (*Shooting someone in the head in a game is deeply satisfying*). | .91 |
| 18. Disfruto compitiendo con otros. (*I enjoy competing with others*). | .50 | 22. Disfruto de las peleas y luchas violentas en el juego. (*I enjoy the violent fights in video games*). | .85 |
| **Cognitive development** | Alpha = .81 | **Customization** | Alpha = .88 |
| | Retest = .69** | | Retest = .68** |
| 3. Me hacen pensar/calentarme la cabeza. (*Games make me think*). | .88 | 7. Disfruto diseñando cosas en el juego. (*I enjoy customizing things in games*). | .95 |
| 11. Me suponen un reto mental. (*Games imply a mental challenge*). | .66 | 15. Me gusta crear cosas en el juego, como casas u otras construcciones. (*I like making things in video games, like houses or other constructions*). | .85 |
| 19. Me hacen más inteligente (*Games make me smarter*). | .36 | 23. Me gusta crear mi propio mundo en el juego. (*I like to create my own world in games*). | .64 |
| **Coping** | Alpha = .87 | **Fantasy** | Alpha = .82 |
| | Retest = .69** | | Retest = .68** |
| 4. Alivia mi estrés. (*It helps me get rid of stress*). | .81 | 8. Disfruto metiéndome en la piel de un nuevo personaje en cada juego. (*I enjoy putting myself into a new character's shoes in each game*). | .66 |
| 12. Me ayuda a mejorar mi estado de ánimo. (*Gaming helps me improve my mood*). | .76 | 16. Me gusta sentirme parte de una historia. (*I like feeling that I'm part of a story*). | .62 |
| 20. Me permite sentirme mejor cuando estoy frustrado. (*Gaming allows me to feel better when I am frustrated*). | .71 | 24. Me siento inmerso en un mundo fantástico/ficticio. (*I feel immersed in a fantastic/fictitious world*). | .59 |

every item were greater than .50 except for Item 19 that had a factor loading of .36. Furthermore, the internal consistency of each motive was good to excellent with Cronbach alphas ranging from .76 to .93. The test-retest scores were acceptable regarding the low number of items per dimension, with all the values higher than .60 [71].

**Criterion validity.** Results showed that VMQ motives had significant correlations with various gaming-related indicators (see Table 3). All motives were associated with the number of hours spent gaming weekly and disordered use, although the motives that presented higher associations were social interaction, coping, and competition. Significant relationships were also found among gaming genres. For instance, according to those genres most consumed, shooters were mainly associated with violent reward, social interaction and competition, whereas playing MOBA games and sport games were strongly related to competition. All the motives were associated with male gender except customization. The largest associations among males were observed with competition, violent reward, and social interaction. Descriptive data by gender and correlations between motives are displayed in S2 and S3 Tables.

Multiple linear regression analyses were conducted to identify the role of motives in number of hours spent gaming weekly and disordered use, controlling for gender and age (see Table 4). Higher social interaction and coping predicted greater gaming use, whereas coping, social interaction and, to a lesser extent, violent reward predicted disordered videogame use.

## Study 3. Young adult sample study

### Method

**Participants and procedure.** The convenience young adult sample that participated in the survey was comprised of 260 undergraduate participants who played at least one hour a week (41.9% males) with a mean age of 20.53 years ($SD$ = 3.63). The survey was selflessly completed online via *Google Forms*.

**Measures.** *Videogaming Motives Questionnaire (VMQ)*. The VMQ administered to adolescents was used in order to examine its psychometric properties as well as to confirm the eight-factor solution in the young adult sample. Please see Study 2 Methods section for details about the VMQ item structure.

*Gaming behaviors*. As in the adolescent sample (Study 2), participants reported the number of weekly hours spent gaming, online and offline, and provided up to five of their most played videogames. Videogame genres were classified in the same way as in Study 2.

*Internet Gaming Disorder Test (IGD-20 Test)-Spanish versión*. The Spanish version of the IGD-20 test [72] was used to assess disordered gaming in young adults. The scale includes 20 items answered using a 5-point Likert scale from 1 (*strongly disagree*) to 5 (*strongly agree*). The IGD-20 Test assesses disordered gaming activity during the past 12 months using the nine criteria for IGD. The Cronbach's alpha was very good in the present study (.86).

**Statistical analysis.** A CFA was performed on the VMQ testing the eight-factor model with robust methods using the EQS software, version 6.1 [73]. The model's goodness-of-fit was assessed by employing the following fit indices: Satorra-Bentler chi-squared (S−B$\chi^2$), Satorra-Bentler normed chi-squared (S−B$\chi^2$/d.f.), the comparative fit index (CFI), the incremental fit index (IFI), the non-normed fit index (NNFI), and the root mean square error of approximation (RMSEA). For a good model fit, the fit indices must meet the following values: S−B$\chi^2$ (non-significant), S−B$\chi^2$/df (between 1 and 2), CFI, IFI, and NNFI ($\geq$ .95), and RMSEA $\leq$ .05 [74]. Last, as the $\chi^2$ is very sensitive to the model complexity or sample size, we used the S−B$\chi^2$/df to overcome this problem [74].

As with the adolescent sample (Study 2), the correlations between the VMQ subscales and the number of weekly hours spent gaming, disordered gaming, and gaming genre were

**Table 3. Correlation analysis between motivations of the VMQ and other relevant variables for Studies 2 and 3.**

| Study 2 Adolescent sample | Age | Gender | Gaming hours | Disordered gaming | Shooter (197) | MOBA (71) | Strategy (64) | MMORPG (3) | Role-playing (23) | Action-adventure (45) | Sport (150) | Casual (33) | Social simulation (35) | Construction (21) | Platform (23) | Fighting (20) |
|---|---|---|---|---|---|---|---|---|---|---|---|---|---|---|---|---|
| Recreation | .00 | -.17** | .23** | .18** | .23** | .00 | -.08 | .05 | .15** | .07 | .04 | -.06 | -.02 | .10 | -.10 | .06 |
| Competition | .03 | -.46** | .32** | .39** | .32** | .20** | -.02 | .03 | -.06 | -.02 | .25** | -.12* | -.26** | -.04 | -.18** | .07 |
| Cognitive development | .05 | -.20** | .24** | .37** | .21** | .12* | .06 | .06 | .07 | .19** | .03 | .03 | -.15** | .03 | -.08 | -.03 |
| Coping | .00 | -.18** | .34** | .48** | .20** | .03 | -.05 | .04 | .10 | .14** | .10 | -.05 | -.15** | .04 | -.11* | .08 |
| Social interaction | .02 | -.34** | .37** | .51** | .40** | .15** | -.13* | .09 | -.02 | -.03 | .06 | -.03 | -.20** | -.03 | -.17** | -.02 |
| Violent reward | -.01 | -.42** | .28** | .36** | .47** | .09 | -.15** | .00 | -.09 | -.02 | .15** | -.19** | -.25** | .00 | -.22** | .10 |
| Customization | -.01 | .06 | .19** | .18** | .08 | .04 | .04 | .08 | .09 | .12* | -.12* | .07 | .27** | .18** | -.04 | .01 |
| Fantasy | -.05 | -.15** | .26** | .31** | .20** | .06 | -.05 | .14** | .18** | .22** | -.03 | -.01 | .04 | .12* | -.07 | -.01 |
| **Study 3 Young adult sample** | **Age** | **Gender** | **Gaming hours** | **Disordered gaming** | **Shooter (77)** | **MOBA (74)** | **Strategy (46)** | **MMORPG (11)** | **Role-playing (61)** | **Action-adventure (25)** | **Sport (33)** | **Casual (51)** | **Social simulation (23)** | **Construction (8)** | **Platform (13)** | **Fighting (10)** |
| Recreation | .14* | -.30** | .29** | .29** | .26** | .19** | -.05 | .15* | .23** | .20** | .12 | -.18** | .13* | .02 | .04 | .07 |
| Competition | .00 | -.36** | .15* | .28** | .12 | .18** | .03 | .13* | -.04 | -.03 | .19** | -.26** | -.12 | .02 | -.08 | .10 |
| Cognitive development | .19** | -.18** | .12* | .21** | .13* | .06 | .03 | .04 | .06 | .17** | .01 | .08 | -.03 | -.02 | -.03 | .08 |
| Coping | .15* | -.16* | .24** | .39** | .18** | .06 | -.10 | .18** | .11 | .22** | .11 | -.11 | -.02 | .00 | -.07 | .04 |
| Social interaction | -.02 | -.40** | .42** | .41** | .30** | .43** | -.05 | .26** | .17** | .04 | .02 | -.31** | -.16** | .05 | -.09 | .12 |
| Violent reward | -.06 | -.46** | .21** | .43** | .37** | .11 | -.14* | .08 | .14* | .24** | .18** | -.34** | -.11 | -.03 | .09 | .13* |
| Customization | .06 | .09 | .15* | .10 | .13* | -.04 | -.04 | .08 | .20** | .07 | -.04 | -.16** | .35** | .12 | -.05 | -.02 |
| Fantasy | .04 | -.17* | .24** | .34** | .25** | .07 | -.09 | .18** | .30** | .32** | .06 | -.29** | .13* | .01 | .11 | .09 |

*Note.*

\* *p* < .05

\*\* *p* < .01.

Gender: 1 = males, 2 = females. The number of players by genre are indicated between parentheses.

**Table 4. Multiple linear regressions of hours spent gaming and disordered gaming for Studies 2 and 3.**

| | | Study 2 | | Study 3 | |
| --- | --- | --- | --- | --- | --- |
| | | Adolescent sample | | Young adult sample | |
| | | Gaming hours | Disordered gaming | Gaming hours | Disordered gaming |
| 1 | Gender | -.28*** | -.26*** | -.32*** | -.39** |
| | Age | -.03 | .02 | -.04 | -.07 |
| | $\Delta R^2$ | .08*** | .07*** | .10*** | .15*** |
| 2 | Recreation | .05 | -.08 | .13 | .00 |
| | Competition | .02 | .02 | -.11 | -.01 |
| | Cognitive development | -.06 | .04 | -.08 | -.03 |
| | Coping | .18** | .30*** | .05 | .21** |
| | Social interaction | .19** | .29*** | .33*** | .19** |
| | Violent reward | .06 | .12* | -.01 | .18* |
| | Customization | .09 | .00 | .03 | -.15* |
| | Fantasy | -.02 | -.03 | .03 | .17* |
| | $\Delta R^2$ | .13*** | .27*** | .13*** | .18*** |
| | $R^2$ | .21 | .34 | .23 | .33 |

*Note*. 1 = males, 2 = females. β = standardized beta; $\Delta R^2$ = change in variance; $R^2$ = total $R^2$.

*$p < .05$.

**$p < .01$.

***$p < .001$.

calculated. Finally, in order to test the role of gaming motives in the number of hours spent gaming weekly and disordered gaming, hierarchical multiple regression analyses were conducted, controlling for age and gender.

**Ethics.** As in Study 1, participants provided written informed consent and were informed about the study, confidentiality and treatment of data, as well as the data protection procedure. The study was approved by the ethical committee from the Universitat Jaume I and was conducted in accordance with the Declaration of Helsinki.

## Results

**Gaming behavior.** The young adult sample played an average of 9.26 hours a week (*SD* = 11.97). The videogames played reported by participants (n = 250) were classified according to gaming genre: shooter (n = 89), MOBA (80), casual (n = 71), role-playing (n = 70), strategy (n = 57), sports (n = 40), social simulation (n = 33), action-adventure (n = 28), platform (n = 17), MMORPGs (n = 11), fighting (n = 11), construction (n = 9) and other miscellaneous games (n = 33).

**Factor structure.** Results from the CFA demonstrated a good model fit (S–B$\chi^2$ = 330.13, *df* = 224, $p < .001$; S–B$\chi^2$/df = 1.47; CFI = .97; IFI = .97; NNFI = .96; RMSEA = .04), although the chi-square test did not acquire non-significance. This result from the chi-square test was expected due to the index's sensitivity to sample size. The loadings ranged from .62 to .95 (see Fig 1); and 81.9% of the variance was explained by the 8-factor model.

**Criterion validity.** As with the adolescent sample (Study 2), VMQ motives demonstrated significant correlations with other relevant variables in gaming (see Table 3). The highest correlations between motivation scores and number of hours spent gaming weekly were for social interaction and recreation. In relation to disordered gaming, the highest correlations were with violent reward, social interaction, and coping. Regarding gaming genre, compared to the

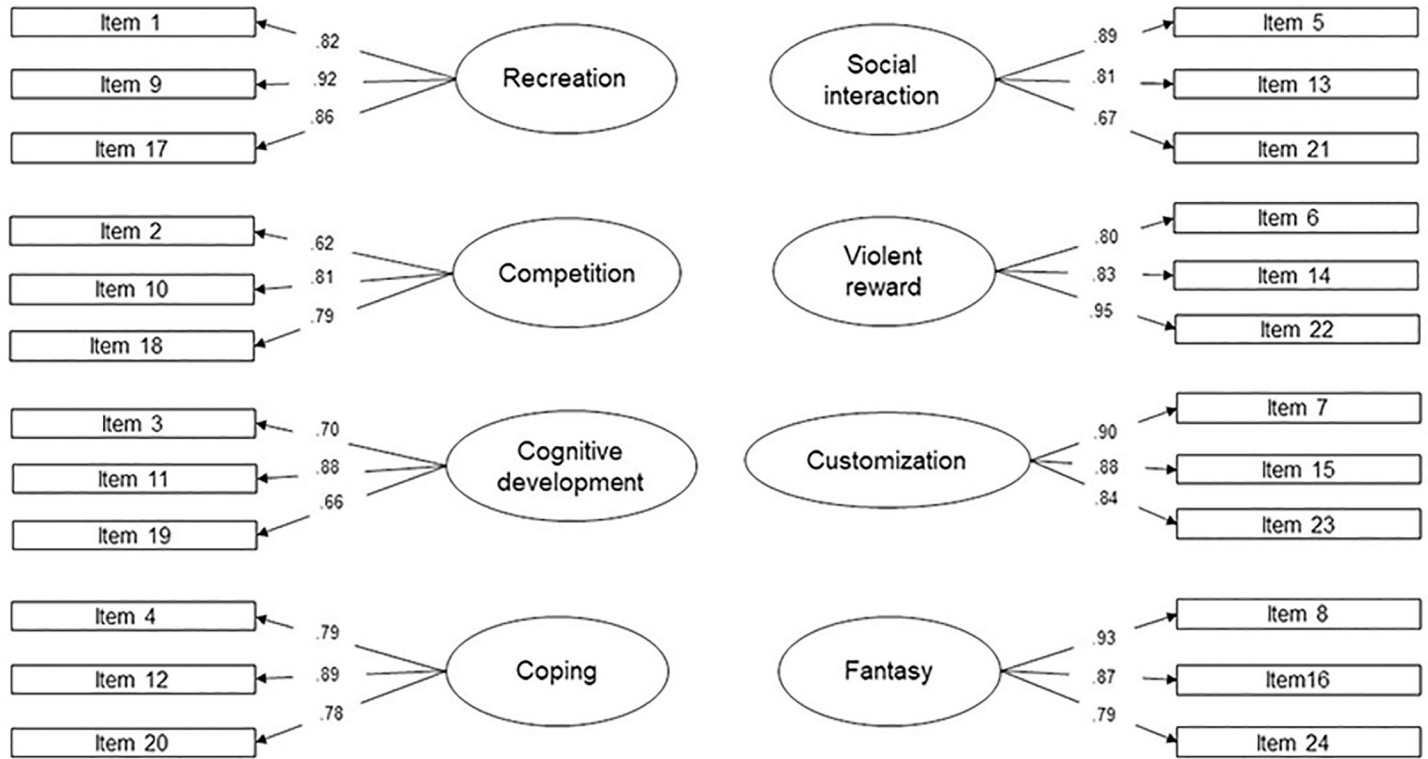

**Fig 1. Confirmatory factor analysis of the VMQ among the young adult sample.** For clarity, covariances between motives and errors were not depicted in the figure. All factor loadings were significant at $p < .001$.

adolescent sample, similar findings were found among the most popular genres. Shooter games were strongly associated with violent reward and social interaction, and MOBA games were highly related to social interaction. As in the adolescent sample, male gender showed relationships with every motive except customization. The largest associations among males were violent reward, social interaction, and competition. In addition, customization was broadly associated with higher age. Descriptive data by gender and correlations between motives are displayed in S1–S3 Tables.

Multiple linear regression analyses were carried out to identify the role of motives in number of hours spent gaming weekly and disordered gaming, controlling for age and gender (see Table 4). Results showed that social interaction was associated with higher gaming use whereas coping and, to a lesser extent, social interaction and violent reward mainly predicted disordered gaming.

## Discussion

The aim of the present study was to develop a new psychometrically robust scale for assessing motives for videogame playing, the Videogaming Motives Questionnaire (VMQ). As a result, the VMQ consisted of an 8-intercorrelated-factor model that appeared to best explain the data. It showed good internal consistency and temporal stability reliability indices, and also presented evidence of validity when predicting video game frequency of use and disordered gaming.

Motives are important antecedents of other addictive-related behavior such as using alcohol, tobacco, marijuana and gambling (e.g., [27,63]). Based on the motivational model of Cox

and Klinger [75], three main and common motives to all these addictive behaviors have been identified: social, enhancement, and coping [22,25,27]. A fourth motive of conformity (drinking alcohol or smoking cigarettes in response to social pressures) has also been proposed [22,25], although its role in substance use has usually been small or even negligible [63,76]. Accordingly, the main three common motives, coping, social, and enhancement (also labeled as *diversion* or *recreation*), have been identified in previous studies on videogame playing (see Table 1), and in the present study.

The coping motivation refers to playing for stress-reduction and mood enhancement. This motivation is widely represented in most gaming motive scales that include an escape/coping motivational component. Demetrovics et al. [36] through a literature review, distinguished between escapism (playing to forget daily problems) and coping motives. However, these motives were highly correlated. $r > .6$; [36,49] and, in the present study, both motives converged as a unique factor with coping items loading more strongly. In the current study, the coping motive was associated with shooter and action-adventure games both in adolescents and young adults, and with MMORPGs in young adults. Likewise, previous research has shown the escape/coping motivational component as being highly related to MMORPGs among adults [47,59].

In the present study, coping was the strongest and somewhat selective predictor of disordered gaming scores among both adults and adolescents. This result is in accordance with previous studies in which escape/coping motives presented the highest association with disordered gaming [48–50,52,53,59,77] and with problematic substance use and gambling [22–27]. This motive appears to be especially relevant in the presence of psychological problems and environmental stressors, such as depression and anxiety symptoms [50,54–56] or stressful life events [78]. Accordingly, motivation for escape or relief from negative mood constitutes an important criterion for the diagnoses of IGD [5] and other addictions in the DSM-5 [3].

The social interaction motive is based on bonding with friends and making new ones. This motive is also widely represented in most of the scales. In the present study, social interaction was broadly associated with shooters and MOBAs both in adolescents and young adults, and MMORPGs among young adults, whereas the motive was negatively associated with games that are usually played alone, such as social simulation games (for both adolescents and young adults). Similarly, social interaction has been associated with higher preferences for online role-playing, shooters, and real time strategy games [47] and lower engagement in casual games [57]. In the present study, social interaction was the highest predictor of number of hours spent gaming weekly among both samples, in line with other studies (e.g., [32,49,53,57,79]). Therefore, more socially motivated players like to spend their time playing videogames that offer social interactions such as MOBAs or/and shooters.

The present study also found that social interaction was highly associated with disordered gaming (after controlling for age, gender, and the remaining motives in both samples). Similarly, previous studies have found that after escape/coping, the gaming motive with the next highest relationship with disordered gaming is social interaction [51,59]. It has been suggested that high scores in social interaction within gaming might reflect low social competence in real life, and that gaming compensates for this characteristic, resulting in disordered gaming [14,15].

Finally, recreation (enhancement or recreational) motives–focusing on the enjoyable and recreational gaming component–was mainly related to higher use of shooter and role-playing games in both samples. In addition, some studies have found that recreation is the major motivation related to greater gaming duration [32,57]. In the present study, recreation motives were correlated with frequency of gaming and disordered gaming, but when age and gender and other gaming motives were controlled for, these associations became nonsignificant,

probably due to the interrelation with other motives. Therefore, in our samples, other motives such as social interaction may explain better the variance in gaming frequency rather than playing for mere enjoyment.

In addition to these three common motives, specific motivational factors have also been identified for other addictive behaviors, such as expanse motives for marijuana use and abuse [24,25] or financial motives for gambling behavior [26]. In the field of gaming, previous studies have also identified specific motives for playing videogames, such as arousal (playing to excite emotions), achievement-challenge (playing to obtain in-game rewards or prestige), skill development, fantasy, competition, and customization. In addition, Hilgard et al. [59] identified two additional relevant motives for playing–violence catharsis (in-game violence helping to release negative moods or aggression) and violent reward–which were included in the present study due to the potential relevance of aggressive gaming in violent behaviors [62]. Of these specific motives, the study clearly identified skill development (cognitive development in the present study), fantasy, competition, customization, and violent reward. In Study 1, items from arousal motives mainly loaded on the recreation factor, violent catharsis items loaded on the escape-coping factor, and achievement-challenge items were distributed between different motives.

The *cognitive development* motive refers to the intellectual activity stimulation during videogame play. In the first study, this motive encompassed two previous motives: skill development [36] and 'smarty-pants' [44], although when selecting the items with higher loadings for the final VMQ, it became a more cognitive development and mental challenge factor. In the present study, this motive was associated with action-adventure and shooter games, in both samples, and with MOBA games among adolescents. Previous research has found that shooters and real time strategy genres have been highly associated with skill development [47]. Consequently, individuals with higher levels in this motive may be more attracted to those games where improving skill and abilities are important elements of the in-game experience.

The *fantasy* motive is defined by playing for the immersion in the gaming world and the story's in-game characters. This motive has received different labels in gaming motive research such as immersion [34], presence [31], narrative [33], story [59], or exploration [61]. In the present study, fantasy was significantly associated with game genre use in which a storyline is developed, usually through a campaign, such as action-adventure games, shooter game, role-playing games, and MMORPGs.

Previous research has reported similar findings in which fantasy is strongly associated with the use and preference for shooters, role-playing, and action-adventure games [46,47,57,58] as well as for franchises of such genres [59]. In addition, the present study showed that fantasy significantly predicted higher disordered gaming among young adults, in line with some studies [40,50,54]. Therefore, in-game immersion, at least among adults, could impair the player's life by neglecting important life domains. In other words, this higher immersion feeling in videogame playing may disrupt professional careers and social relationships affecting players' psychological wellbeing.

The *competition* motive–based on the pleasure of competing and winning against others– was strongly associated with competitive gaming genre use such as shooter games (in the adolescent sample), MOBA games and sports games (for both samples). In addition, competition was negatively related to the use of non-competitive genres such as casual games (in both samples). Similarly, the competition motive has been found to be strongly associated with competitive gaming genres, such as sports games, action games, or real time strategy games, and to a lesser extent noncompetitive games and genres such as casual games [47,57,58]. Taken together, these studies highlight that the competition motive appears to play an important role in the player's game genre preferences.

*Customization* motives refers to the creation and design of things in-game. Consequently, this motive was highly associated with use of social simulation games in both samples. Customization was also extensively associated with construction (among adolescents) and role-playing games (among young adults). In previous research, customization has been positively associated with higher preferences for role-playing games such as *Final Fantasy* and *Skyrim* [59]. Among the adult sample in the present study, customization may have acted as a protective factor in preventing disordered gaming. The gratification of creating and designing in-game things might underlie facets of conscientiousness that are negatively associated with disordered gaming [9].

Finally, *violent reward* refers to the gratification obtained via in-game violence. Surprisingly, this motive has been very scarcely studied, despite its potential role in the development of aggressive preferences and violent behavior. Accordingly, this motive was strongly associated with use of shooters and negatively related to non-violent and more peaceful videogames such as casual games, both in adolescents and in young adults. Similarly, Hilgard et al. [59] reported that this motive was strongly associated with preferences for the action shooter franchise of *Grand Theft Auto*. In addition, in the present study, violent reward also had a positive association with disordered gaming among both adolescents and young adults. As far as the present authors are aware, this is the first time that this association has been reported, although other studies have shown that the acceptance of violence is associated with higher disordered gaming [80], and that MMORPG players attracted to the release of aggressive and antisocial feelings through gaming have a higher risk of addiction [81]. Overall, these relationships highlight the importance of this often-neglected motive in particularly harmful videogame behaviors, such as disordered gaming and the preference for violent games.

To sum up, adolescents and young adults displayed a similar pattern in the relationship between motives and gaming behaviors. For both samples, disordered gaming was strongly related to coping, followed by social interaction and violent reward, this last motive with a lower effect. In addition, a larger number of gaming hours was associated with higher scores in social interaction. However, slight differences existed between young adults and adolescents. For instance, disordered gaming was slightly related to fantasy and negatively to customization in young adults, whereas number of hours spent gaming weekly were also associated with coping in adolescents. With regard to videogame genres, a similar pattern was also found, replicating most of the highest associations between motives and games played in both samples.

Regarding gender, adolescent and young adult males presented higher scores than females in nearly all scales apart from customization, especially in competitive, violent reward and social interaction motives. These higher motivational levels in male gender for almost all motives has been systematically found in previous studies (e.g., [32–36,57]). Along these lines, it has been reported that adolescent males prefer competitive and violent game genres, such as action-*shooters*, sport, fight or strategy games; whereas girls prefer game genres such as brain and skill games, and social simulation games, more characterized by customization aspects [9]. These gender-based differences on motives for gaming may help to explain why males, who prefer games featuring more time-consuming and engagement characteristics, presented higher prevalence of regular and disordered use of video games than females [7].

The present study is not without its limitations. First, the studies only comprised Spanish adolescents and young adults. Therefore, validation of the VMQ in other socio-cultural contexts are necessary to generalize the findings. Second, the present study was cross-sectional, consequently longitudinal studies are needed to offer stronger evidence of motives as determinants of gaming behaviors. Third, despite trying to be exhaustive, there could be other relevant motives not identified in the present study. Fourth, the time engaged in gaming was obtained by self-reported retrospective estimates, which have been found to be less accurate than other

procedures such as Timeline Followback [82]. Finally, in order to systematically explain individual differences in gaming behaviors, future studies should include other relevant variables such as personality, which may allow exploration of the complex interplay between these variables (e.g. the mediational role of motives in the personality-gaming associations) [83].

## Conclusion

The Videogaming Motives Questionnaire showed an underlying robust structure comprising eight motives: recreation, social interaction, coping, violent reward, competition, fantasy, cognitive development and customization. In addition, it showed meaningful associations with different game-related behaviors in two independent samples of adolescents and young adults, such as weekly time spent gaming, disordered gaming, and gaming genres most played.

Thus, VMQ constitutes a new psychometric instrument that produces valid and reliable scores for the assessment of some of the more representative gaming motives, and in a relatively brief way. The possibility to assess gaming motives briefly and comprehensively may help to incorporate the study of motives in the research of individual differences on gaming-related behaviors, and may be useful in the development of personalized programs to prevent and treat disordered gaming [84].

## Supporting information

**S1 Table. Study 1: Pattern matrix for oblimin eight-factor solution extracted from pilot Videogaming Motives Questionnaire (VMQ).**
(DOCX)

**S2 Table. Study 2: Descriptive data by gender, gender differences and correlation matrix.**
(DOCX)

**S3 Table. Study 3: Descriptive data by gender, gender differences and correlation matrix.**
(DOCX)

## Acknowledgments

The authors wish to thank all the undergraduate participants, and all the students, parents, and teachers of the high schools *Bovalar* and *El Caminàs* for making this study possible.

## Author Contributions

**Conceptualization:** Francisco J. López-Fernández, Manuel I. Ibáñez.

**Data curation:** Francisco J. López-Fernández.

**Formal analysis:** Francisco J. López-Fernández, Laura Mezquita.

**Funding acquisition:** Generós Ortet, Manuel I. Ibáñez.

**Investigation:** Francisco J. López-Fernández.

**Methodology:** Francisco J. López-Fernández, Laura Mezquita, Manuel I. Ibáñez.

**Project administration:** Generós Ortet, Manuel I. Ibáñez.

**Resources:** Generós Ortet.

**Supervision:** Mark D. Griffiths, Manuel I. Ibáñez.

**Writing – original draft:** Francisco J. López-Fernández.

**Writing – review & editing:** Laura Mezquita, Mark D. Griffiths, Generós Ortet, Manuel I. Ibáñez.

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
