## [Decision Letter · Decision Letter 0]

30 Jul 2020

PONE-D-20-12358

The development and validation of the Videogaming Motives Questionnaire (VMQ)

PLOS ONE

Dear Dr. Ibáñez,

Thank you for submitting your manuscript to PLOS ONE. After careful consideration, we feel that it has merit but does not fully meet PLOS ONE’s publication criteria as it currently stands. Therefore, we invite you to submit a revised version of the manuscript that addresses the points raised during the review process.

We look forward to receiving your revised manuscript.

Kind regards,

Francesca Chiesi

Academic Editor

PLOS ONE

Journal Requirements:

2. Please include additional information about your scoping review in your methods section, including search strategy, inclusion and exclusion criteria, to enable reproducibility and replicability.

3. Please proofread your manuscript for typos (for instance item 3 of table 2).

4. Your ethics statement must appear in the Methods section of your manuscript. If your ethics statement is written in any section besides the Methods, please move it to the Methods section and delete it from any other section. Please also ensure that your ethics statement is included in your manuscript, as the ethics section of your online submission will not be published alongside your manuscript.

Reviewers' comments:

Reviewer's Responses to Questions

**Comments to the Author**

1. Is the manuscript technically sound, and do the data support the conclusions?

Reviewer #1: Yes

2. Has the statistical analysis been performed appropriately and rigorously? 

Reviewer #1: Yes

3. Have the authors made all data underlying the findings in their manuscript fully available?

Reviewer #1: Yes

4. Is the manuscript presented in an intelligible fashion and written in standard English?

Reviewer #1: Yes

5. Review Comments to the Author

Reviewer #1: Ms #: PONE-D-20-12358

Title: The development and validation of the Videogaming Motives Questionnaire (VMQ)

Authors: López F, et al.

Summary:

This study examined the factor structure of a new instrument, the Videogaming Motives Questionnaire, in three samples of Spanish adolescents and young adults. Study 1 used exploratory factor analysis (EFA) to define empirical factors in young adults. The EFA revealed that an 8-factor model best explained the data. This pattern was replicated in Study 2 in a sample of adolescents. Finally, Study 3 used confirmatory factor analysis to demonstrate a selective mapping/loading of items to the 8 respective motive factors. Overall psychometric properties indicated clear simple structure and internal reliability of all factors. The sample size was sufficient to conclude the models are stable. Coping was identified as a key motive linked with gaming duration and with disordered gaming (DG), based on an author-compiled index of DG. Social interaction was also strongly linked with heavier gaming and DG. The latter result may reflect a preference for on-line socializing by people who are not entirely comfortable with direct interpersonal encounters. Violent motives also coincided with DG, while male gender coincided with stronger endorsement of all motives, aside from Customization, and DG, consistent with other addictive behaviors. The Discussion interpreted the findings for each of the key motives and proposed possible reasons for the pattern of effects.

Comments/Questions:

1. Ms p. 3, line 51

The term "gaming addiction" is used along with a number of others (disordered gaming, pathological video gaming, disordered gaming behavior) to describe the syndrome under investigation. For simplicity, it would be helpful if the authors select a single term to describe the syndrome and use that consistently throughout the manuscript. They might even mention, parenthetically, that they will be using this term to describe the target syndrome, the first time it is introduced.

2. Ms p. 7 - line 103

The term "IGD" is introduced but not defined. Perhaps Internet Gaming Disorder (IGD) is the term they wish to use as their standard description of the syndrome?

3. Table 1 provides a very clear summary of the findings to date.

4. Ms p. 8, lines 129-130:

"All these deficiencies impede..." It is not clear that these are "deficiencies" as opposed to different conceptual or explanatory frameworks or preferred terminology. Perhaps a more neutral description (e.g., "This variability across the different gaming scales impedes...") would work equally well?

5. In the Introduction, it would be helpful to specify the age of “adolescents” (12-17 years)? "young adults" (18-25 years)?

6. Page 8 - When describing the Participants, it would be helpful to report any incentive offered or alternative reason to participate in the study, or if none was offered, this could be stated as well. Same for studies 2 and 3.

6. Ms p. 9, line 151 "recruited from the LAN party" The term LAN should be defined for general readership

7. Ms p. 9, line 156 "Through an exhaustive scoping review of the literature"

It would be helpful to specify which databases were examined, the years of publication deemed eligible for inclusion, and a sample of the search terms used.

8. Ms p 9, line 172

What was the rationale for using oblique as opposed to orthogonal rotation to achieve simple structure in the two EFAs?

9. Ms p 11, lines 220-2

Weekly number of gaming hours was obtained by adding online and offline daily hours and multiplying the days of these periods of the week by the total number of hours. Global retrospective estimates of time engaged in activities like drinking or gambling have been found to be less accurate than values based on a Timeline Followback (see Sobell and Sobell 1992) procedure. Since Timeline Followback was not mentioned here, the limitations of the current reporting procedure should be acknowledged in the Discussion

10. Ms p 12, line 246

Hierarchical multiple regressions were used to examine the relationship between motives, gaming intensity and problems, while controlling for age and gender. What was the method of entry of predictors in each stage, and briefly what was the rationale for this?

11. Ms p. 18

Study 3 refers to the Emerging Adulthood sample. This is the only time this terminology is used. Were the age inclusion criteria different than for study 2? If not, perhaps it is best to stick with 'Young Adult' as the description?

12. Ms p. 21

In the Discussion, before launching into the specific motives that emerged, it would be good to state that an 8-factor model appeared to best explain the data in all 3 samples, and to report the proportion of variance explained by these factors in each case. The first paragraph under Conclusion, p 26 addresses many of these points and could be moved up to the beginning of this section.

Some comment on the inter-correlation vs. orthogonality of factors might also be made and what this overall pattern of scores means for VMQ as assessment tool, generally. The consistent elevation in motive endorsement by males, aside from customization, may also merit some comment.

13. As the authors note, Coping emerged as a reliable and somewhat selective predictor of disordered gaming (as compared with Social Interaction, for example). Although this makes intuitive sense, I think this linkage, which, again as noted, is seen with other addictive behaviors, might be considered a bit more with respect to addiction pathology.

For example, what is the difference between wanting to engage in a behavior and needing to do so?

How do these different global motivations correspond to the concept of "dependence" within the addiction framework?

And what would be the expected role of environmental factors like "stress" as a conditioned (i.e., involuntary) cue for gaming in people with these different motivational profiles?

Minor Points:

1. Ms p. 4, line 71 "based on well-stablished theories" Typo: well-[e]stablished

2. Ms p. 7, line 119 "the described motives suffer from [?] of ‘jingle-jangle’ fallacies," Missing words [a form]

3. Table 2, Cognitive Development factor: Item 3 (Games make me thing). Typo - Games make me thin[k]

4. Ms p. 19, line 320 "we used the S−Bχ2/df to overcome such problem [74]." Syntax "to overcome [this] problem"

6. PLOS authors have the option to publish the peer review history of their article (what does this mean?). If published, this will include your full peer review and any attached files.

Reviewer #1: No

---

## [Author Response · Author response to Decision Letter 0]

11 Sep 2020

Journal Requirements:

Response: Done

2. Please include additional information about your scoping review in your methods section, including search strategy, inclusion and exclusion criteria, to enable reproducibility and replicability.

Response: Dear editor, 

We regret to inform you that there was a translation misunderstanding (an example of “lost in translation”) with regard to the term “scoping review”. We conducted an extensive review of motivational scale studies, selecting most of the relevant questionnaires used in research, but it was not a scoping review. Later, when the English language revision of the paper was carried out, “extensive review” was changed to “scoping review”. This was unfortunate as these concepts are different. We would like to apologize again. Accordingly, we have changed the term “scoping review” for the original “extensive review”. In any case, we have described in more detail which databases were examined, the years of publication eligible for inclusion, and a sample of the search terms used, as suggested (see p. 9, lines 160-165)

3. Please proofread your manuscript for typos (for instance item 3 of table 2).

Response: Done

4. Your ethics statement must appear in the Methods section of your manuscript. If your ethics statement is written in any section besides the Methods, please move it to the Methods section and delete it from any other section. Please also ensure that your ethics statement is included in your manuscript, as the ethics section of your online submission will not be published alongside your manuscript.

Response: Done

Response to reviewer:

We really appreciate the reviewer’s contribution. 

1. Ms p. 3, line 51

The term "gaming addiction" is used along with a number of others (disordered gaming, pathological video gaming, disordered gaming behavior) to describe the syndrome under investigation. For simplicity, it would be helpful if the authors select a single term to describe the syndrome and use that consistently throughout the manuscript. They might even mention, parenthetically, that they will be using this term to describe the target syndrome, the first time it is introduced.

Response: p. 3, line 53-54 

“These inclusions in the DSM-5 and ICD-11 assume that disordered gaming, (for simplicity, this term will be used throughout the manuscript to describe the syndrome)… “

2. Ms p. 7 - line 103

The term "IGD" is introduced but not defined. Perhaps Internet Gaming Disorder (IGD) is the term they wish to use as their standard description of the syndrome?

Response: p. 3, line 47-49

… defined as a “persistent and recurrent use of the Internet to engage in games, often with other players, leading to clinically significant impairment or distress ...”

3. Table 1 provides a very clear summary of the findings to date.

4. Ms p. 8, lines 129-130:

"All these deficiencies impede..." It is not clear that these are "deficiencies" as opposed to different conceptual or explanatory frameworks or preferred terminology. Perhaps a more neutral description (e.g., "This variability across the different gaming scales impedes...") would work equally well?

Response: p. 8, lines 132-133

“This variability across the different gaming scales impedes…”

5. In the Introduction, it would be helpful to specify the age of “adolescents” (12-17 years)? "young adults" (18-25 years)?

Response: p. 3, line 43

“… adolescents (12-17 years) and young adults (18-24 year)”

6. Page 8 - When describing the Participants, it would be helpful to report any incentive offered or alternative reason to participate in the study, or if none was offered, this could be stated as well. Same for studies 2 and 3.

Study 1

Response: p. 9, line 156

“Participants voluntarily completed an online survey without any compensation”

Study 2

Response: p. 11, line 210

“… and received school supplies as an incentive”

Study 3

Response: p. 19, line 323

“The survey was selflessly completed online”

6. Ms p. 9, line 151 "recruited from the LAN party" The term LAN should be defined for general readership

Response: pp. 8-9, lines 153-154

“.. an event at the Jaume I University in which gamers gathered to play multiplayer video games, ..”

7. Ms p. 9, line 156 "Through an exhaustive scoping review of the literature"

It would be helpful to specify which databases were examined, the years of publication deemed eligible for inclusion, and a sample of the search terms used.

Response: p. 9, lines 160-165

Through an extensive review of the literature concerning gaming motivation scales, 25 relevant gaming motive scales were selected. For the review, it was conducted a search in the databases PsycINFO and Google Scholar from 2000 to present, combining the key words motives or motivation, video game or gaming, scale or questionnaire. Furthermore, citations from selected studies of our database were used to continue the review process until no other relevant study was found.

8. Ms p 9, line 172

What was the rationale for using oblique as opposed to orthogonal rotation to achieve simple structure in the two EFAs?

Response: 

Oblimin rotation (which does not assume prior relations between factors and allows axes to freely rotate) is somewhat more exploratory than orthogonal rotation (which assumes independence between factors and forces axes to be orthogonal), and also allows to estimate the intercorrelations between factors. Based on previous psychometric studies on motives for gaming and other addictive behaviors, we expected high intercorrelations between factors. Accordingly, oblimin rotation was used in the pilot study. The pilot study confirmed that factors were highly intercorrelated, so the oblimin rotation was also used in the EFA of the second study, and the correlations between factors were specified in the CFA of the third study.

Thus, we have added in Statistical Analysis section of study 2:

p. 13, lines 255-257 “As the pilot study showed that motives were intercorrelated, an EFA with principal axis factoring and oblique rotation (oblimin) was performed for testing the VMQ with original eight-factor model using IBM SPSS Statistics V21.0.”

9. Ms p 11, lines 220-2

Weekly number of gaming hours was obtained by adding online and offline daily hours and multiplying the days of these periods of the week by the total number of hours. Global retrospective estimates of time engaged in activities like drinking or gambling have been found to be less accurate than values based on a Timeline Followback (see Sobell and Sobell 1992) procedure. Since Timeline Followback was not mentioned here, the limitations of the current reporting procedure should be acknowledged in the Response: Discussion

p. 29, lines 562-564

“Fourth, the time engaged in gaming was obtained by self-report retrospective estimates, which have been found to be less accurate than other procedures such as Timeline Followback [82]”.

10. Ms p 12, line 246

Hierarchical multiple regressions were used to examine the relationship between motives, gaming intensity and problems, while controlling for age and gender. What was the method of entry of predictors in each stage, and briefly what was the

rationale for this?

Response: p. 13, lines 263-267

“Previous research has shown that gender and age were relevant variables in video game–related behaviors, such as disordered use or video game genre preferences (e.g., [9,32,57]). Therefore, hierarchical multiple regressions were controlled for gender and age by introducing them in step 1, whereas motives were entered in step 2”.

11. Ms p. 18

Study 3 refers to the Emerging Adulthood sample. This is the only time this terminology is used. Were the age inclusion criteria different than for study 2? If not, perhaps it is best to stick with 'Young Adult' as the description?

Response: p. 19, line 318

“Study 3. Young adult sample study”

12. Ms p. 21

In the Discussion, before launching into the specific motives that emerged, it would be good to state that an 8-factor model appeared to best explain the data in all 3 samples, and to report the proportion of variance explained by these factors in each case. 

AND

The first paragraph under Conclusion, p 26 addresses many of these points and could be moved up to the beginning of this section. 

AND

Some comment on the inter-correlation vs. orthogonality of factors might also be made and what this overall pattern of scores means for VMQ as assessment tool, generally. 

AND 

The consistent elevation in motive endorsement by males, aside from customization, may also merit some comment.

Response: p. 22., lines 402-405

“As a result, the VMQ consisted of an 8-intercorrelated-factor model that appeared to best explain the data. It showed good internal consistency and temporal stability reliability indices, and also presented evidence of validity when predicting video game frequency of use and disordered gaming.” 

And the information of % of variance is now presented at the Result sections of each study (Factor Structure) 

Initial testing: 78.7%; Adolescent sample study: 78.5% & Young adulthood sample: 81.9%

p. 28, lines 546-556

“Regarding gender, adolescent and young adult males presented higher scores than females in nearly all scales apart from customization, especially in competitive, violent reward and social interaction motives. These higher motivational levels in male gender for almost all motives has been systematically found in previous studies (e.g., [32-36, 57]). Along these lines, it has been reported that adolescent males prefer competitive and violent game genres, such as action-shooters, sport, fight or strategy games; whereas girls prefer game genres such as brain and skill games, and social simulation games, more characterized by customization aspects [9]. These gender-based differences on motives for gaming may help to explain why males, who prefer games featuring more time-consuming and engagement characteristics, presented higher prevalence of regular and disordered use of video games than females [7].”

13. As the authors note, Coping emerged as a reliable and somewhat selective predictor of disordered gaming (as compared with Social Interaction, for example). Although this makes intuitive sense, I think this linkage, which, again as noted, is seen with other addictive behaviors, might be considered a bit more with respect to addiction pathology.

For example, what is the difference between wanting to engage in a behavior and needing to do so?

How do these different global motivations correspond to the concept of "dependence" within the addiction framework? And what would be the expected role of environmental factors like "stress" as a conditioned (i.e., involuntary) cue for gaming in people with these different motivational profiles?

Response: We have tried to reflect all these very insightful considerations in the following paragraph: 

p. 23, lines 426-434

“In the present study, coping was the strongest and somewhat selective predictor of disordered gaming scores among both adults and adolescents. This result is in accordance with previous studies in which escape/coping motives presented the highest association with disordered gaming [48-50,52,53,59,77] and with problematic substance use and gambling [22-27]. This motive appears to be especially relevant in the presence of psychological problems and environmental stressors, such as depression and anxiety symptoms [50, 54-56] or stressful life events [78]. Accordingly, motivation for escape or relief from negative mood constitutes an important criterion for the diagnoses of IGD [5] and other addictions in the DSM-5 [3] 

Minor Points:

Done

---

## [Editor Report · Decision Letter 1]

23 Sep 2020

PONE-D-20-12358R1

The development and validation of the Videogaming Motives Questionnaire (VMQ)

PLOS ONE

Dear Dr. Ibáñez,

Thank you for submitting your manuscript to PLOS ONE. After careful consideration, we feel that it has merit but does not fully meet PLOS ONE’s publication criteria as it currently stands. Therefore, we invite you to submit a revised version of the manuscript that addresses the issues detailed below (see Additional Editor Comments).

We look forward to receiving your revised manuscript.

Kind regards,

Francesca Chiesi

Academic Editor

PLOS ONE

Additional Editor Comments:

Dear Authors:

I appreciate the strength of the paper and I believe that the changes made in response to the reviewer’s comments contributed to improve it. However, I have a minor comment about the scales you used to measure disordered gaming. You employed two different scales in Study 2 and Study 3, so I believe that this point deserves some explanation. Could you motivate your choice? Did this choice have an impact on your results? Additionally, was the Gentile’s scale validated in Spanish (as the IGD-20 )? If not, how do you translate it?  

---

## [Author Response · Author response to Decision Letter 1]

29 Sep 2020

Additional Editor Comments:

Dear Authors:

I appreciate the strength of the paper and I believe that the changes made in response to the reviewer’s comments contributed to improve it. However, I have a minor comment about the scales you used to measure disordered gaming. You employed two different scales in Study 2 and Study 3, so I believe that this point deserves some explanation. Could you motivate your choice? Did this choice have an impact on your results? Additionally, was the Gentile’s scale validated in Spanish (as the IGD-20 )? If not, how do you translate it? 

Response: 

Dear Editor:

Thanks for your comments.

We employed different scales to measure disordered gaming in each study because of the sample’s age. The scale used in the adolescent sample of Study 2 was originally developed for the assessment of gaming disorders in youths aged from 8 to 18 years (see Gentile, 2009), and validated in Spain with an adolescent sample (the mean age of the validation sample in Spain was 14,29 years old; see López-Fernández et al., 2020). In the sample of young adults of Study 3, we considered a better option to use the IGD-20, originally developed in a sample of young adults (Pontes et al., 2014) and validated two years later in Spain (Fuster et al., 2016) (mean age of the original sample of 26.5 years old, and 26.14 years old in in the Spanish sample).

Although these scales somewhat different, both were based on similar DSM criteria, so we consider that this difference in the assessment of gaming disorder would have a limited impact on the results. This idea has been reinforced by the fact that disordered gaming assessed with the different scales have converged in finding that the most important motives for disordered gaming have been coping, social interaction and violent reward (in the same order). In any case, we have added a new limitation:

“Five, two different scales was employed for assessing disordered gaming in adolescents and young adults. Although we have found that main motives related to disordered gaming were the same in both samples, some of the differences found between adolescents and young adults could be attributed to the different scales used.”

Last, the Gentile’s scale usen in study 2 was validated in Spanish adolescents in a previous study (López-Fernández et al., 2020). Extracted from López-Fernández et al., 2020: 

“The original 11 items was back-translated… For the study, the participants indicated their frequency of video game-related problems on a 4-point Likert scale (from 0 = “never or almost never” to 3 “almost always or always”) during the last 12 months. According to the parallel analysis run using Monte Carlo PCA (Watkins, 2006), a one-factor structure was obtained with the EFA, where all items presented adequate factor loadings ranging from .49 to .81. Cronbach’s alpha in this sample was .88”.

Thus, now we have included this fact in the text as follows:

p. 13, lines 248-249

“A Spanish adaptation [9] of a measure especially developed to assess disordered gaming in adolescents [70] was employed”.

---

## [Editor Report · Decision Letter 2]

2 Oct 2020

The development and validation of the Videogaming Motives Questionnaire (VMQ)

PONE-D-20-12358R2

Dear Dr. Ibáñez,

We’re pleased to inform you that your manuscript has been judged scientifically suitable for publication and will be formally accepted for publication once it meets all outstanding technical requirements.

Kind regards,

Francesca Chiesi

Academic Editor

PLOS ONE

---

## [Editor Report · Acceptance letter]

15 Oct 2020

PONE-D-20-12358R2 

The development and validation of the Videogaming Motives Questionnaire (VMQ) 

Dear Dr. Ibáñez:

I'm pleased to inform you that your manuscript has been deemed suitable for publication in PLOS ONE. Congratulations! Your manuscript is now with our production department. 

Kind regards, 

on behalf of

Dr. Francesca Chiesi 

Academic Editor

PLOS ONE